# Detecting and Understanding Sentiment Trends and Emotion Patterns of Twitter Users—A Study on the Demise of a Bollywood Celebrity

**Ahmed Al Marouf** [1,*] , **Jon G. Rokne** [1] and **Reda Alhajj** [1,2,3]

1 Department of Computer Science, University of Calgary, Calgary, AB T2N 1N4, Canada
2 Department of Computer Engineering, Istanbul Medipol University, Istanbul 34810, Turkey
3 Department of Heath Informatics, University of Southern Denmark, 5230 Odense, Denmark
* Correspondence: ahmedal.marouf@ucalgary.ca

**Abstract:** Detecting societal sentiment trends and emotion patterns is of great interest. Due to the time-varying nature of these patterns and trends this detection can be a challenging task. In this paper, the emotion patterns and trends are detected among social media users in a certain case and it is noted that the detection of the trends and patterns is especially difficult in this medium because of the use of informal language. In particular, the role of social networks in the expression of emotions relating to the death of a well-known and loved Bollywood actor Sushant Singh Rajput (SSR) by their fans is explored. The data for the analysis of the emotional state and the sentiment levels of the fans has been acquired from Twitter posts. Different existing sentiment analysis algorithms were compared for the study and chosen for identifying the sentiment trend over a specific timeline of events. The same Twitter posts were also analyzed for emotional content by extracting linguistic features using the psycholinguistic package, Linguistic Inquiry and the Word Count package (LIWC), relating to emotions. Additionally, viral hashtags extracted from the Twitter posts have been segmented and analyzed in order to identify new viral hashtags expressed by the posts over time. The associations between the old and new viral hashtags and between sentiment trends and emotional shifts among the fan base of SSR have been determined and presented graphically.

**Keywords:** social media; sentiment trend; emotion pattern; Sushant Singh Rajput (SSR)



## 1. Introduction

Social media platforms (e.g., Facebook, Twitter, Instagram etc.) have become the main source of information for many of its users. Many of the users are also using these platforms for other tasks such as communication between friends/connections, branding online shops, creating news, sharing opinions and so on. At the inception of online social media, the platforms were simple and not very sophisticated. The platforms have, however, improved and been enhanced over time so that in addition to the social network function they have become an important vehicle for interacting with other users by means of texts, photos and chats generating large amounts of data of various kinds. This user-generated content (UGC) can be mined for user profile-based analysis and detection systems, such as personality detection [1], recommendation systems [2,3], sentiment analysis [4], sarcasm detection [5], etc.

Currently according to the most recent statistics published by Zephoria [6], a digital marketing company, the Facebook metric Family Monthly Active People (MAP) shows Facebook had 3.51 billion users as of 28 July 2021, an increase of 12 percent year-over-year [6]. Twitter, the other heavily used social media platform, had increased their year-over-year monetizable Daily Active Users (mDAU) by 34 percent to 186 million mDAUs worldwide in 2020 [7]. Because of this increase in active users, the diversity in terms of culture, gender, and age of users is also expected to increase.

Users of social media platforms post discussions and comments on a variety of topics. Often, there is an expression of emotion underlying these topics. Identifying and extracting these emotions is a cross-domain research topic involving computer science, psychology, and neurobiology [8]. The emotions contained in the social media posts are expressed in various ways, such as using textual expression, emoticons, meme posts or related images. One of most widely used contents are text messages which can be analysed using the advanced natural language processing tools. Some of these text messages, including Facebook and Twitter text postings are therefore a potential source of textual data that can be interpreted as being emotional by using the insights provided by the word usage patterns. Interpretation of this use of vocabulary is part of psycholinguistic research and analyzing the textual data is part of natural language processing. The structure of social media networks is another important aspect to be considered while analyzing the posts.

The textual data extracted from posts on social media platforms such as Facebook, Twitter etc. are considered to be user-generated content [9]. Such user generated textual data in the posts is very much informal. The textual data contains deep insights into personality traits, which can be extracted after first performing a great deal of pre-processing and then using psycholinguistic methods [10].

The main motivation for this paper is to try to understand the psychology, behind posts and hashtags generated by the fan base of Bollywood actors in terms of sentiment and emotion. To further elucidate the aim of the paper, we have not worked on the psychological aspects or traits of the social media users in particular, but rather on the expressive responses by them. In particular, the news of the suicide of Sushant Singh Rajput together with the subsequent news suspecting that he might have been murdered has generated a large volume of social media posts having a particular emotional pattern. Indeed, this case did turn into a huge controversy in the world of Bollywood, the largest movie industry in the world. Based on the timeline of the events published by the news and paper media, people reacted differently to each event.

The aim of this paper is therefore to analyze sentiment trends and emotion patterns toward films and actors expressed by fan bases. It is the psychological viewpoint of the Twitter posts over time that is used to identify emotional patterns of the fan base. A second aim is to analyze the usage of hashtags by the social media users who are fans of SSR, who as noted, passed away in 2020. The analysis also includes how new hashtags have evolved due to the new media reports that has led to a sentiment that something should be done to honor the departed soul. Finding starting hashtags and associated hashtags that have gone viral over time and developing insights from the new hashtags that have appeared among users is another focus of the paper.

## 2. Event of Study

The term "Bollywood" is a portmanteau of "Bombay" and "Hollywood" indicating the local geographic and artistic origins of this industry which has now spread to all of India. According to Statistica, the total revenue of Bollywood in the year 2021 was around 93 billion dollars [11].

There is a huge fan base for Bollywood that uses social media platforms, especially Twitter, for expressing their reviews on Bollywood films, acting styles of Bollywood actors, quality of their acting etc. The fans also react to other Bollywood industry activities such as trends in Twitter claiming extreme levels of nepotism, trends related to the #MeToo movement, discussions on the demise of Sushant Singh Rajput and the subsequent CBI investigations, the marriage of Katrina Kaif and Vicky Kausal, the marriage of Alia Bhatt and Ranbir Kapoor etc. Accusations of the industry being controlled by political parties are also discussed, due to the involvement of 3 leaders from the Bharatiya Janata Party (BJP), the current ruling party of India, in the controversies. The Economist published an article on "BJP vs. Bollywood" saying "... as cases of COVID-19 were rising in India's first wave and the BJP was preparing for elections in the poor eastern state of Bihar, a young actor called

Sushant Singh Rajput (SSR)—a Bihari—committed suicide in Mumbai" [12]. Therefore, the demise of SSR does not only affect the fan base, but also the political establishment.

The significant events which took place after the demise of SSR are illustrated in Figure 1. Starting from finding the dead body until the confirmation of suicide by the supreme court, the timeline shows the behavior change of Twitter users based on these significant events.

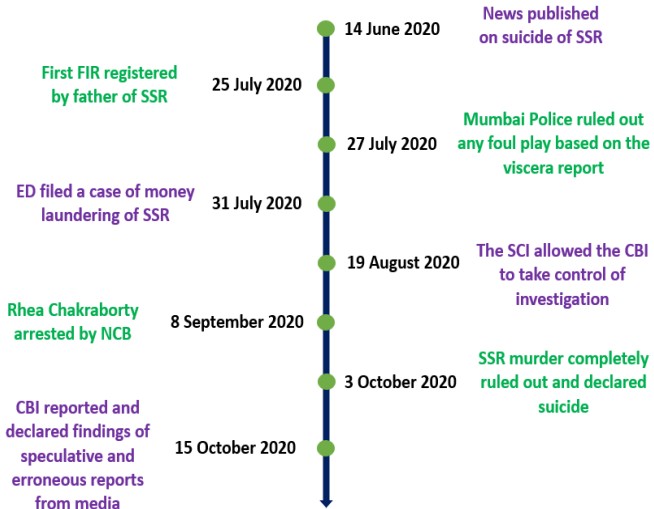

**Figure 1.** Timeline of significant events related to SSR case. [SSR: Sushant Singh Rajput; FIR: First Information Report; ED: Enforcement Directorate; SCI: Supreme Court of India; CBI: Central Bureau of Investigation; NCB: Narcotics Control Bureau].

Viral posts or hashtags by the fan base of their favorite celebrity, Sushant Singh Rajput, the very popular Bollywood actor, contain in some cases, extreme emotional expressions when posting on the event of their death. In this paper, we therefore want to identify the emotional state developed among Twitter users after their death. The news of their death was published on 14 June 2020, and the report stated it was a suicide. Following this, their fan base became so affected by the incident that they started showing what can be interpreted as extreme levels of emotions through Twitter posts.

The death of Sushant Singh Rajput (SSR) caused the fan base to become surprisingly emotional, mainly because he used to be a very open-minded actor and also a bit of a geek and Twitter users created viral hashtags such as #SushantSinghRajput, #SSR, #SSRSuicide etc. when discussing their death, while the newspapers and TV media first reported their passing as a suicide case, subsequently, some new evidence appeared during media investigations indicating that their death might possibly be a murder case. Because of this information, the fans reacted fiercely. A new hashtag was created for postings discussing the investigation of the case as if it was a case of murder. Evidence such as doubtful behavior of their last girlfriend and their friends was noted, and fans highlighted those viewpoints in Twitter posts. The government of India reopened the case of their death and proceeded to have the case being re-investigated by the Central Bureau of Investigation (CBI) and Narcotics Control Bureau (NCB). After transferring the case from the police to CBI, the case got more complex day by day as more evidence was forthcoming regarding the suspicion of murder. As time passed, the emotions and the tone of the language of posts discussing the SSR case evolved as expressed by their linguistic content and new hashtags.

## 3. Related Works

Posting information about current events such as COVID-19 [13], information about US Elections [14] and other events has become a normal activity for many social media users nowadays. Frequently these postings go viral, that is, they are re-posted repeatedly by new users as they read the posts. Unfortunately, this re-posting can also include

the spreading of misinformation and/or disinformation [15]. For example, re-posting misguided information about coronavirus infections became a world-wide issue once it became an epidemic. Similarly, during the elections certain politicians are spreading false "alternative truths" [16]. In the case of the coronavirus, among the resulting societal impact of social media misinformation includes panic buying [17] and rumor spreading rumor [18]. This shows that the social media users are exhibiting very diversified as well as adverse patterns caused by the uncertainties associated with the ongoing development of the epidemic.

A recent paper surveyed on social media postings relating to COVID-19 covered Twitter emotion research in detail [19]. The paper covered the evolution of emotion descriptions and emotion models for textual posts in general. The different methods that are covered include keyword-based, lexicon-based, machine learning and hybrid methods. Methods for emotion intensity detection, sarcasm detection and emotion-cause detection are also summarized in the paper.

Posts or hashtags going viral or becoming trended can happen within a very short period of time over multiple social media. For example, posts by politicians tend to go viral before elections [20] and post about celebrities are frequently going viral when they perform publicity stunts [21]. Interestingly, viral posts often contain multiple hashtags which leads towards using hashtags to map the propagation of posts through networks.

In the literature, case studies have been done on suicide related publicity, mainly focusing on how TV or newspaper media react to such events. A meta-analysis was presented by T. Niederkrotenthaler et al. [22] on a celebrity suicide case reflected in media coverage. A pilot study has been reported in a medical journal on cross-country comparison of media reporting of celebrity suicide in the immediate week following the passing of the celebrity [23]. The effect of media coverage on suicide cases and the quality of media reports are presented in [24]. None of these studies have focused on the use of social media data for analyzing these cases. This paper, therefore, fills a gap in the literature through the analysis of emotions generated from posts relating to a suspected suicide.

The main contributions of the paper are:

- A novel approach to the use of celebrity suicide-related tweets to extract psycholinguistic features from textual information, and to determine sentiments and emotion patterns among the posters of the tweets.
- Timeline based analysis to determine new viral hashtags based on relationships among SSR related hashtags.
- Detection and analysis of sentiments and emotions from SSR related tweets.

## 4. Research Methodology

The research methodology described in this paper that is applied to the set of Twitter posts considered has several steps which are described in detail in the next subsections.

### 4.1. Proposed Framework

In this section, the proposed framework and its implementation is discussed including data collection from Twitter, data pre-processing, annotation and a timeline-based approach to detect and understand the sentiment trend and emotion pattern expressed by the collected data.

#### 4.1.1. Conceptual Diagram of the Framework

The conceptual diagram of the proposed framework to detect the sentiment trend and emotion pattern is shown in Figure 2.

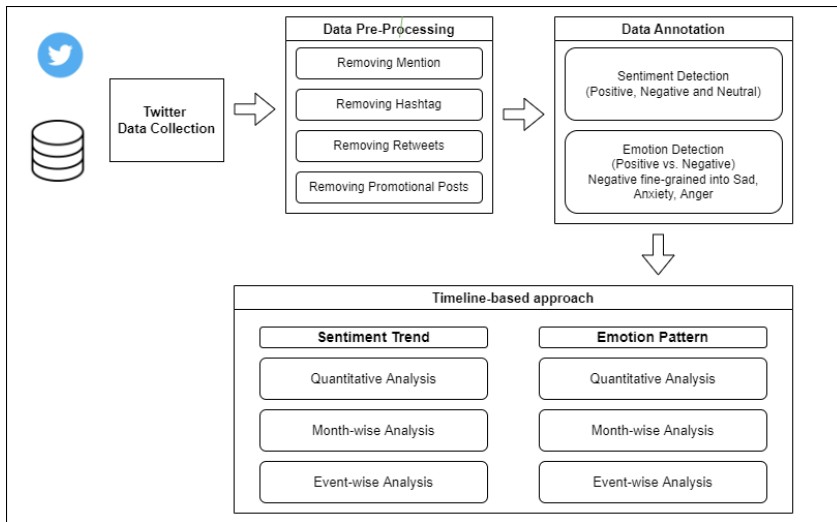

**Figure 2.** Overview of proposed framework for detecting sentiment trend and emotion pattern.

For this framework all the data of the corpus is collected from Twitter and then fetched into the pre-processing step to remove erroneous and extraneous elements from the corpus. The collected text corpus contains mentions, hashtags, retweets and some promotional posts as they are commonly found in the Twitter posts. After removing those, the pre-processed data is used for annotating sentiment and emotion. Finally, quantitative, month-wise and event-wise analyses were performed to understand and visualize the trend and pattern in sentiment and emotion.

### 4.1.2. Data Collection

There are a number of ways that the data from Twitter posts can be collected, such as manually, or using Twitter API. Manual data collection is a time-consuming process while the Twitter API provides the data which are only publicly available. If data-scraping software is used, then this tends to be costly. On the other hand, using the Twitter API [25] only limits the data to the most recent data (30 days). Based on post-related information, such as hashtag, geographic location, topics of post etc. it gives integrated information (i.e., number of retweets, number of replies, web link etc.) for each of the posts. Table 1 shows the statistics of collected raw data from twitter. In total 16,647 posts have been collected from total 1214 unique Twitter users having four different hashtags (#SSR, #JusticeForSSR, #WorldUnitedForSSR and #FeedFood4SSR).

It was therefore decided to use the Python package, GetOldTweets version-3 [26] for the data collection. The package is appropriate for extracting the old posts based on the location or timeline of the tweets. The relevant data items extracted for each of the Twitter posts are date of tweet, username, no. of replies, no. of retweets, no. of favorites, text of the tweet, mentions, hashtags, id and the permanent link of the post. Figure 3 shows the steps of data collection process from Twitter.

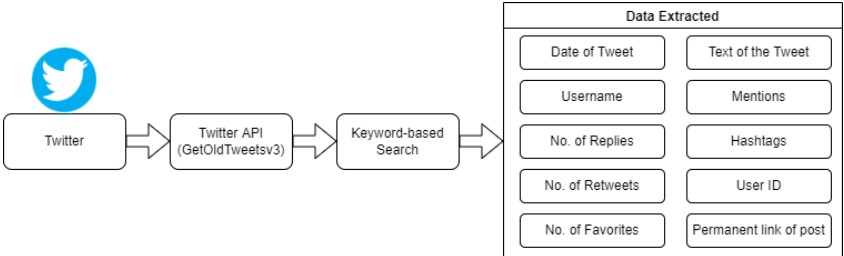

**Figure 3.** Process of data collection from Twitter.

The date information was collected since it was to be used for timeline-based representation of the tweets and for finding associated emotion pattern for each tweet. The difference between emotion patterns and sentiment patterns over time can also be visualized using this data.

**Table 1.** Statistics of data collected from Twitter

| Properties | Values |
|:---:|:---:|
| No. of Unique Users | 1214 |
| Total No. of Posts | 16,647 |
| No. of Posts using #SSR | 9500 |
| No. of Posts using #JusticeForSSR | 6200 |
| No. of Posts using #WorldUnitedforSSR | 879 |
| No. of Posts using #FeedFood4SSR | 68 |

4.1.3. Data Pre-Processing

This section describes the pre-processing steps applied to the collected data. For pre-processing textual data from social media, several aspects, such as mentions, hashtags, retweets and/or promotional posts need to be considered. For this purpose, the Python regular expression (re) package [27] and substitute function was used to segment and remove Mentions, Hashtags, Retweets and Promotional Posts. The reasons for removing these parts were:

- Mentions are formatted as @USERNAME which mentions someone from the same social media that are included in this post. Using such usernames would mean that the privacy policies of the social media platforms would be violated, which is why these parts were skipped.
- Hashtags "#HASHTAG" are used to attract other users and making the post relative to any practical matters. Therefore, using these hashtags to extract the psycholinguistic features may cause problems for extracting the emotional content of the posts.
- Retweets (RT) are used for replying to someone else posts. These retweets are themselves considered to be tweets containing a good textual source for the purpose of the analysis of emotions. The word retweet has therefore been removed from the posts and the posts were accepted.
- Promotional Posts are made using viral hashtags to attract customers who engage with the hashtags. Such posts are not related to the hashtag context. Instead, they are simply promotions. Promotional posts generally contain "http", "https" or "https://" and "html tags". Therefore, if any of these parts were detected in a post, then the post was removed by using regular expression functions.

After the pre-processing the statistics for the data are shown in Table 2. In total 15,566 posts remain after data cleaning steps performed as pre-processing, coming from 1087 unique users. ¶

After segmenting these parts as a data pre-processing step, the processed data was fed into the LIWC system for feature extraction. Some of the posts were removed using the above criteria and 15,566 Twitter posts remained that could be further analyzed.

*4.2. Sentiment Detection and Analysis Approach*

To determine sentiment patterns from a set of Twitter posts, the sentiments of each post must be recognized. The overall flow of analysis for the detection and analysis of the sentiments is shown in Figure 4.

**Table 2.** Statistics of the dataset after pre-processing.

| Properties | Values |
|---|---|
| No. of Unique Users | 1087 |
| Total No. of Posts | 15,566 |
| No. of Posts using #SSR | 8827 |
| No. of Posts using #JusticeForSSR | 5897 |
| No. of Posts using #WorldUnitedforSSR | 782 |
| No. of Posts using #FeedFood4SSR | 60 |

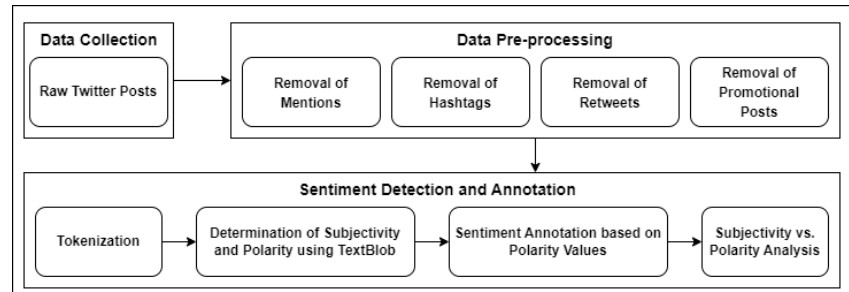

**Figure 4.** Overview of proposed mechanism for sentiment detection and analysis.

In more detail, the data is first collected from Twitter. Then, data pre-processing steps, such as removal of mentions, retweets and promotional posts are performed. After the pre-processing step, the posts are tokenized, where by tokenization [28] is meant the splitting of collections of words, for example sentences, into units that are amenable to sentiment analysis. Determining the subjectivity and polarity value of a given text is then done using the Textblob package [29], which is a python package that enables the determination of the subjectivity and polarity values of a given text. The text could either be a single sentence or a whole a paragraph. Word-by-word polarity values are then utilized to determine a final polarity value for a given text fragment. From this polarity value the sentiment class (i.e., positive, negative and neutral) is determined and annotated.

The pseudocode in Table 3 shows the steps of sentiment analysis and annotation based on the polarity values. The dataframe for the pseudocode consists of the pre-processed Twitter posts, which are already tokenized. Then in the pseudocode the dataframe is passed to the subjectivity and polarity functions and a score is generated for each Twitter post. Based on the polarity score the getAnnotation() function is applied which determines the sentiment class labels. This function is used to annotate the whole dataframe. The subjectivity values are not used to determine the sentiment label, though they are used to understand the quality of the dataset through the subjectivity vs. polarity scatter plot. The scatter plot is illustrated and described in Section 6.1.

For analyzing the sentiment of tweets, Python NLTK [30] and Textblob [29] packages, which are widely accepted by the research community, were used. These packages were used to estimates the strengths of positive and negative values of emotions extracted from short informal texts [31,32]. As the tweet texts are considered to be short texts due to the character limitation imposed by the Twitter authority; this sentiment score extracting tool was found to be appropriate for finding the sentiments of the tweets. Based on the sentiment scores a mathematical model (shown in Pseudocode in Table 3) was derived to find the exact sentiment of the overall posts.

**Table 3.** Pseudocode 1—Sentiment Detection and Annotation.

| Require: n > 0, where n is number be Twitter post |
| --- |
| 1. Initialize TextBlob |
| 2. df = "Pre-processed Twitter posts" |
| 3. getSubjectivity(df): |
| return TextBlob(text).sentiment.subjectivity |
| 4. getPolarity(df): |
| return TextBlob(text).sentiment.polarity |
| 5. df['Subjectivity'] = df['text'].apply(getSubjectivity) |
| 6. df['Polarity'] = df['text'].apply(getPolarity) |
| 7. getAnnotation(score): |
| 8. if score < 0 then |
| 9. return 'Negative' |
| 10. else |
| 11. if score == 0 then |
| 12. return 'Neutral' |
| 13. else |
| 14. return 'Positive' |
| 15. end if |
| 16. end if |
| 17. df['Class'] = df['Polarity'].apply(getAnnotation) |

### 4.3. Emotion Detection and Analysis Method

In this section, an overview of the emotion detection and annotation method used is provided. Figure 5 shows the processing architecture. The data collection and data pre-processing steps were exactly the same as for the emotion detection and analysis and the same dataset was used. After the pre-processing had been performed, the pre-processed data was forwarded to the feature extraction step. The feature extraction step is in this instant considerably different from the other text feature retrieval processes in this paper. The regular feature extractors, such as, TF-IDF [33], n-gram models [34] etc. are not used. These feature extractors are used to represent the structure of the sentences inputted, not any kind of emotion or sentiment related relations. Therefore, the use of these methods would not be appropriate in this case. Hence, the LIWC [35] is used to extract the psycholinguistic features, such as positive, negative emotion scores, from the texts.

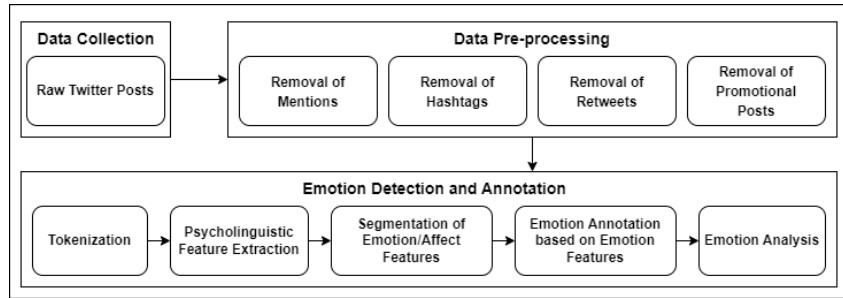

**Figure 5.** Overview of proposed mechanism for emotion detection and analysis.

Psycholinguistic features are numerical or word-counter based features that convey an understanding of the emotional impact of the vocabulary used in a user-generated post. The insights of the post were made clear through the optimal use of the categorization methods proposed in LIWC.

In this step the LIWC (Linguistic Inquiry and Word Count) LIWC 2015 [35], the gold standard for computerized text analysis, was used. This tool can convert the words used in everyday language for expressing our thoughts, feelings, personality, and motivations into categories representing emotions. An earlier version of the program was labelled LIWC2007. The new version, which was used here, provides additional improved features for the analysis. Based on years of scientific research, LIWC2015 is more accurate, easier to

use, and provides a broader range of social and psychological insights compared to earlier LIWC versions. The LIWC2015 outputs 90 attributes from different types of text inputs, such as personal writing, personal email correspondence, professional correspondence, social media posts (Twitter, Facebook, Blog), commercial writing, professional and scientific writings. The core of the text analysis strategy is the LIWC2015 Dictionary. The words and the percentages of positive, negative emotions etc. features acquired (shown in Pseudocode in Table 4) from LIWC2015 were used to annotate the Twitter posts.

The psycholinguistic features that are extracted from the post, which provides the context for natural language processing, are important for understanding the psychological insight of the posts. Table 4, which shows the process of extracting psycho-linguistic features. Then, the labeled data was used for identifying tweet patterns using several machine learning algorithms.

**Table 4.** Pseudocode 2—Emotion Detection and Annotation.

| Require: n > 0, where n is number be Twitter post |
| --- |
| 1. Initialize LIWC and Tokenizer |
| 2. df = "Pre-processed Twitter posts" |
| 3. df tokens = tokenize (dataframe) |
| 4. Import counter from collections |
| 5. category counter = Counter (category for token in dataframe tokens for category in parse(token)) |
| 6. Affect Category Counter = Segment emotional/affect category features only |
| 7. getAnnotation(Affect Category Counter): |
| 8. switch max (Affect Category Counter) do |
| 9. case PositiveEmotion: |
| 10. return 'Positive' |
| 11. case NegativeEmotion: |
| 12. return 'Negative' |
| 13. case Anxiety: |
| 14. return 'Anxiety' |
| 15. case Anger: |
| 16. return 'Anger' |
| 17. case Sad: |
| 18. return 'Sad' |
| 19. df['Class'] = df['Affect Category Counter'].apply(getAnnotation) |

In Figure 6, some features from LIWC have been applied to the annotated data. Only emotion related features were considered, and annotations were applied to find the emotion pattern of the social media entities. Figure 7 shows a snapshot of the annotated data.

For an example, suppose the word "cried" is part of five word-categories: sadness, negative emotion, overall affect, verbs, and past focus. Then Table 5 shows the six (6) LIWC2015 emotions related to the emotion/affect categories of cried.

| | text | WC | Analytic | Clout | Authentic | Tone | WPS | Sixltr | Dic | function | pronoun |
|---|---|---|---|---|---|---|---|---|---|---|---|
| 2 | That doesn't even make any sense. Whatever proof u have should be put either in public | 40 | 32.01 | 59.88 | 15.86 | 2.89 | 13.33 | 15 | 77.5 | 47.5 | 15 |
| 3 | Sushant was murdered ..I repeat Sushant was murdered.We all know It was a planned r | 50 | 17.56 | 88.48 | 30.86 | 25.77 | 25 | 20 | 84 | 48 | 20 |
| 4 | Huge respect to  for exposing truth. | 6 | 99 | 50 | 23.51 | 99 | 6 | 33.33 | 83.33 | 33.33 | 0 |
| 5 | Yes and we all have that deep burning desire that we need to manifest today ... Let's hav | 28 | 21.19 | 85.81 | 1 | 99 | 14 | 17.86 | 67.86 | 39.29 | 17.86 |
| 6 | V auspicious day! Such planetary position comes once in 300 years.. all planets in their m | 37 | 98.79 | 50 | 92.7 | 99 | 7.4 | 18.92 | 70.27 | 29.73 | 5.41 |
| 7 | Please watch this video...bollywood so called celebrities are leaving country for detoxifica | 28 | 29.3 | 92.33 | 13.15 | 25.77 | 9.33 | 25 | 85.71 | 42.86 | 14.29 |
| 8 | Everyone wants justice for SSR . I'm a big fan of him Sonchiriya is my favourite film of hin | 24 | 62.04 | 66.17 | 1 | 92.4 | 8 | 29.17 | 83.33 | 54.17 | 25 |
| 9 | Stop ur so called women drama. Culprit is a culprit no matter wht his /her gender is. She | 41 | 22.11 | 99 | 1 | 25.77 | 10.25 | 12.2 | 70.73 | 46.34 | 19.51 |
| 10 | Stages of  1- The Depression Angle 2 - Nepotism Hit Him Hard 3 - Bollywood Mafia Killed | 41 | 93.26 | 59.64 | 1 | 1 | 20.5 | 17.07 | 41.46 | 24.39 | 7.32 |
| 11 | Kangana, right from the movie Gangster. | 6 | 99 | 50 | 23.51 | 25.77 | 6 | 33.33 | 66.67 | 33.33 | 0 |
| 12 | Can you please respond to my DM for God sake? 3 days and my problem are still unhea | 26 | 33.48 | 64.99 | 5.76 | 25.77 | 8.67 | 23.08 | 84.62 | 50 | 15.38 |
| 13 | I couldn't Plant anything 2day 4  But I wanna dedicate this freshly bloomed flower from r | 41 | 22.11 | 76.78 | 14.43 | 99 | 20.5 | 17.07 | 73.17 | 46.34 | 24.39 |
| 14 | DON'T stop talking about Sushant until his Murderers gets the punishment... We want to | 45 | 28.53 | 99 | 1 | 1 | 5.62 | 11.11 | 91.11 | 57.78 | 24.44 |
| 15 | Please leave your phones for few minutes and pray for our dear Sushant... | 13 | 82.82 | 99 | 17.46 | 99 | 13 | 15.38 | 92.31 | 38.46 | 15.38 |
| 16 | Prayer with tears | 3 | 99 | 50 | 1 | 1 | 3 | 0 | 100 | 33.33 | 0 |

**Figure 6.** Snapshot of LIWC features.

| 1 | text | posemo | negemo | anx | anger | sad | Class |
|---|---|---|---|---|---|---|---|
| 2 | That doesn't even make any sense. Whatever proof u have should be put eithe | 0 | 2.5 | 0 | 0 | 0 | Negative |
| 3 | Sushant was murdered ..I repeat Sushant was murdered.We all know It was a | 6 | 6 | 0 | 6 | 0 | Anger |
| 4 | Huge respect to for exposing truth. | 33.33 | 0 | 0 | 0 | 0 | Positive |
| 5 | Yes di and we all have that deep burning desire that we need to manifest today | 7.14 | 0 | 0 | 0 | 0 | Positive |
| 6 | V auspicious day! Such planetary position comes once in 300 years.. all planets | 8.11 | 0 | 0 | 0 | 0 | Positive |
| 7 | Please watch this video...bollywood so called celebrities are leaving country for | 3.57 | 3.57 | 3.57 | 0 | 0 | Anxiety |
| 8 | Everyone wants justice for SSR . I'm a big fan of him Sonchiriya is my favourite f | 4.17 | 0 | 0 | 0 | 0 | Positive |
| 9 | Stages of 1- The Depression Angle 2 - Nepotism Hit Him Hard 3 - Bollywood Ma | 0 | 7.32 | 0 | 4.88 | 2.44 | Anger |
| 10 | Can you please respond to my DM for God sake? 3 days and my problem are s | 3.85 | 3.85 | 0 | 0 | 0 | Negative |
| 11 | I couldn't Plant anything 2day 4 But I wanna dedicate this freshly bloomed flow | 9.76 | 0 | 0 | 0 | 0 | Positive |
| 12 | DON'T stop talking about Sushant until his Murderers gets the punishment... W | 0 | 8.89 | 0 | 8.89 | 0 | Negative |
| 13 | Please leave your phones for few minutes and pray for our dear Sushant... | 15.38 | 0 | 0 | 0 | 0 | Positive |
| 14 | Prayer with tears | 0 | 33.33 | 0 | 0 | 33.33 | Negative |
| 15 | Wahi yaar.. they should speak up somewhere else instead of attacking people | 0 | 4.55 | 0 | 4.55 | 0 | Anger |
| 16 | Sir plzz take action i m seriously concerned about it | 0 | 10 | 0 | 0 | 0 | Negative |

**Figure 7.** Snapshot of Annotated Data.

**Table 5.** Emotion-related Feature Extraction.

| Feature Name | Abbreviation | Examples |
|---|---|---|
| Affective Process | Affect | Happy, cried |
| Positive emotion | Posemo | Love, nice. Sweet |
| Negative Emotion | Negemo | Hurt, ugly, nasty |
| Anxiety | Anx | Worried, fearful |
| Anger | Anger | Hate, kill, annoyed |
| Sadness | Sad | Crying, grief, sad |

The steps to extract the psycholinguistic features and the annotation of emotional state for each dataframe (df) are shown in the pseudocode in Table 3. For this process, the tokenizer and LIWC package we have to be initialized. The dataframe is then fed into the tokenizer, which outputs the tokens. Then, the counter is imported, which is used for counting the words for each of the categories. There are more than 90 categories (i.e., linguistic dimensions, language variables, psychological processes) [36]. The "category counter" holds the frequency values for each of the categories. Then, only the emotional or affect related categories are segmented from the whole list of categories and stored in "Affect Category Counter". Based on the maximum frequency count on the affect categories, the Twitter post is annotated as that specific category. The switch case determines the category based on the maximum value of the Affect Category Counter. Finally, a new column named "Class" is added to the dataframe.

## 5. Instruments Used

Several tools are used for the experiments in detecting sentiments and emotions. These tools include machine learning tools and Python packages. The main packages used in the experiments were TextBlob [29], NLTK [30] and LIWC [35]. The documentations for each of these packages can be found in the Python Package Index (PyPI) [37]. A user interface and the Application Programming Interface (APIs) for academic and commercial versions of LIWC is publicly available by payment [38].

## 6. Experimental Results

This section demonstrates the results of the experiments related to sentiment and emotion detection along with hashtag segmentation and timeline-based analysis.

### 6.1. Sentiment Detection

For analyzing the sentiment of tweets, the Python NLTK [30] and Textblob [29] packages, which are widely accepted by the research community were used. These packages were used to estimate the strengths of positive and negative values of emotions extracted from short informal texts [26,32]. As the tweet texts are considered to be short texts due to

the character limitation imposed by the Twitter authority this sentiment score extracting tool was found to be appropriate for finding the sentiment of the tweets. Based on the sentiment scores a mathematical model (shown in Pseudocode 1) was derived to find the exact sentiment of the overall tweet. The percentage of sentiment class is determined and shown in Table 6.

**Table 6.** Percentages of Sentiment Analysis.

| Hashtags | Positive | Negative | Neutral |
|---|---|---|---|
| #SSR | 23.8% | 38.9% | 37.2% |
| #JusticeForSSR | 17.9% | 32.6% | 49.5% |
| #WorldUnitedForSSR | 17.2% | 36.7% | 46.1% |
| #FeedFood4SSR | 49.2% | 23.7% | 27.1% |
| #Plant4SSR | 51.2% | 19.7% | 29.1% |

It is evident that the negative sentiment is higher for #SSR, #JusticeForSSR and #WorldUnitedForSSR. On the other hand, the positive sentiment is higher when #FeedFood4SSR and #Plant4SSR hashtags were used. The neutral values are significant, because of this is due to the mixture of positive and negative words, which leads to a neutral mode.

Figure 8 shows the word cloud generated by the most frequent words uttered by the Twitter users using the referred hashtags. These words are mostly utilized for the sentiment analysis unless they are cleaned by any data pre-processing steps.

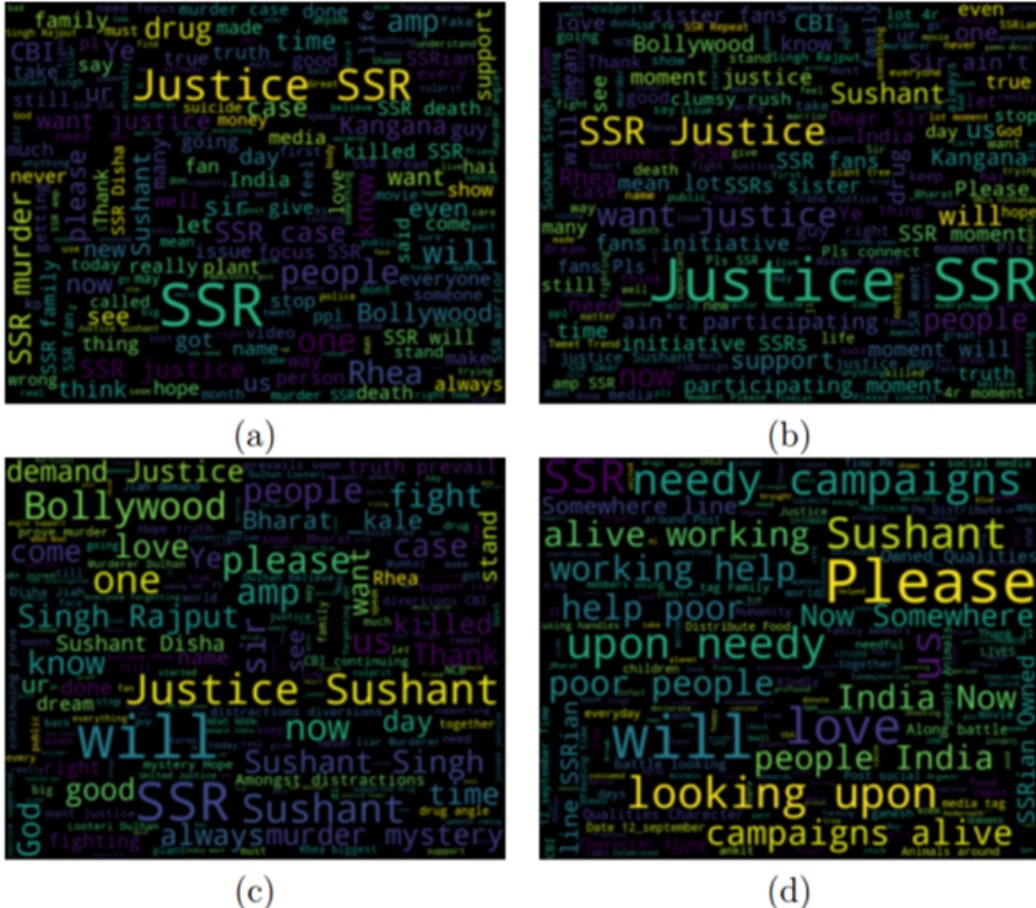

**Figure 8.** Word cloud using the (**a**) #SSR, (**b**) #JusticeForSSR, (**c**) #WorldUnited4SSR, and (**d**) #Food4SSR.

The subjectivity vs. polarity graph is depicted in Figure 9. for #SSR and #JusticeForSSR. From the literatures [39–43], it is evident that the subjectivity vs. polarity graph creates a v-shape that is also found in this study. Basically, the higher scores of subjectivity means

the sentence if more formal, therefore, the polarity scores are also closer to zero, except for some exceptions.

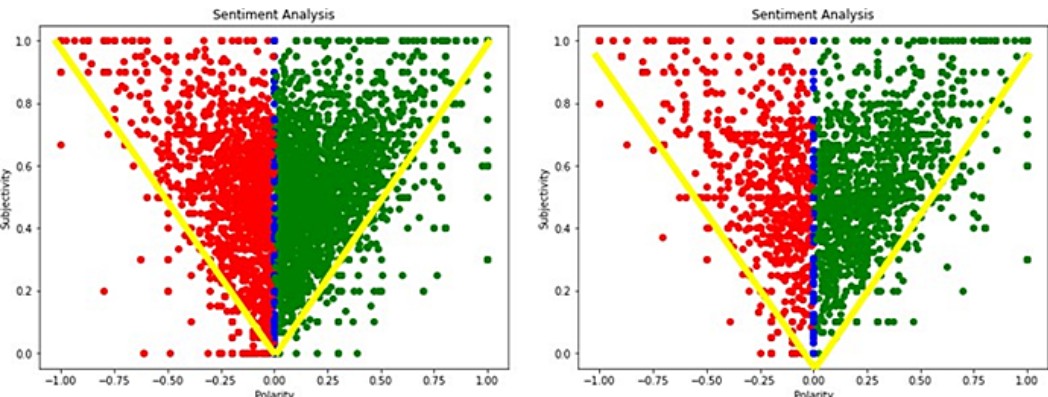

**Figure 9.** Subjectivity vs. Polarity Graph for #SSR and #JusticeForSSR.

### 6.2. Emotion Detection

For the emotion detection and annotation, the psycholinguistic feature extraction step were performed (shown in the Pseudocode in Table 3). The sample of the outcome for the LIWC feature categories is shown in Figure 5. Along with the texts from the dataframe, the numerical values for some of the LIWC features are shown. As we can see, the output is a floating-point number, due to the average weighted counting in the LIWC dictionary. Figure 6 shows the "Class" which is being generated and added to the dataframe (shown in Pseudocode 2). Along with the Affect Category Counter entries, "posemo", "negemo", "anx", "anger" and "sad" are added representing the positive emotion, negative emotion, anxiety, anger and sad values.

Utilizing the psycholinguistic features and annotations, the emotion detection method gives the percentages of emotional pattern for the different hashtags. From Table 7, it is evident that negative emotions are higher than the positive emotion for the first three hashtags (#SSR, #JusticeForSSR and #WorldUnitedForSSR) whereas the positive emotions are higher than the negative emotions for the #FeedFood4SSr and #Plant4SSR hashtags.

**Table 7.** Percentages of Emotion Analysis.

| Tags | Positive | Negative | Anxiety | Anger | Sad |
|------|----------|----------|---------|-------|-----|
| #SSR | 32.4% | 67.6% | 19.3% | 19.1% | 29.2% |
| #JusticeForSSR | 29.8% | 70.2% | 20.3% | 18.2% | 31.7% |
| #WorldUnitedForSSR | 30.9% | 69.1% | 21.5% | 19.5% | 28.1% |
| #FeedFood4SSR | 57.7% | 42.3% | 13.2% | 8.7% | 20.4% |
| #Plant4SSR | 63.1% | 36.9% | 10.8% | 7.2% | 18.9% |

### 6.3. Hashtag Segmentation & Timeline-Based Analysis

A hashtag network graph was generated based on the viral hashtags being used in the posts and the connection between the primary hashtags and the new hashtags was used to identify the trendy hashtags in this special context. The Python NLTK package [30] and Python Regular Expressions (RE) [27] was used to extract regular expressions. The most used hashtags were extracted using the initial hashtags #SSR, #JusticeForSSR, #WorldUnitedForSSR and #FeedFoodforSSR. Using these hashtags, all the associated hashtags used on the same posts were collected. These extracted hashtags were used for generating word clouds. The mostly used hashtags are detected form the word clouds and used as viral hashtags for the further analysis. In Figures 10 and 11, timelines starting from 13 July 2020, the day of the SSR demise, are shown and new hashtags are first detected in August 2020.

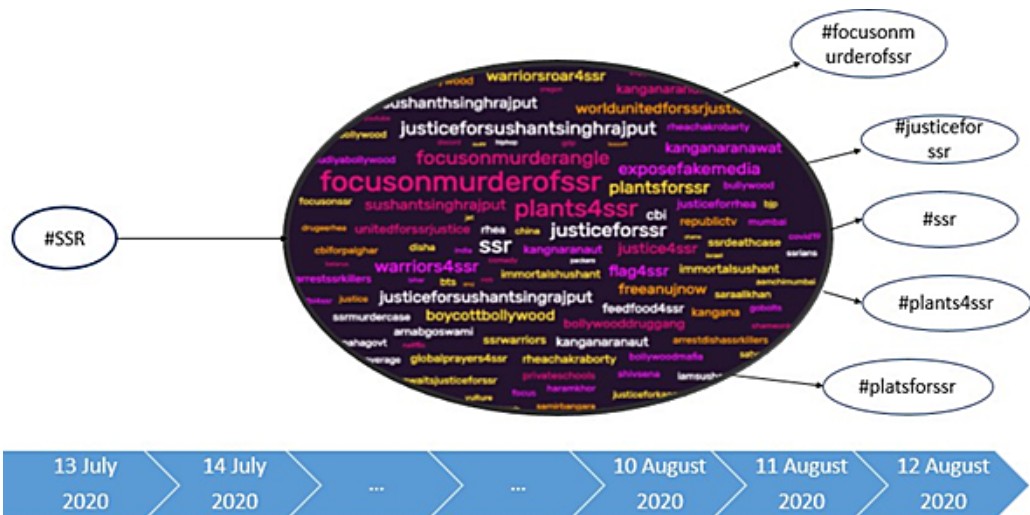

**Figure 10.** Viral Hashtags detection using initial hashtag #SSR.

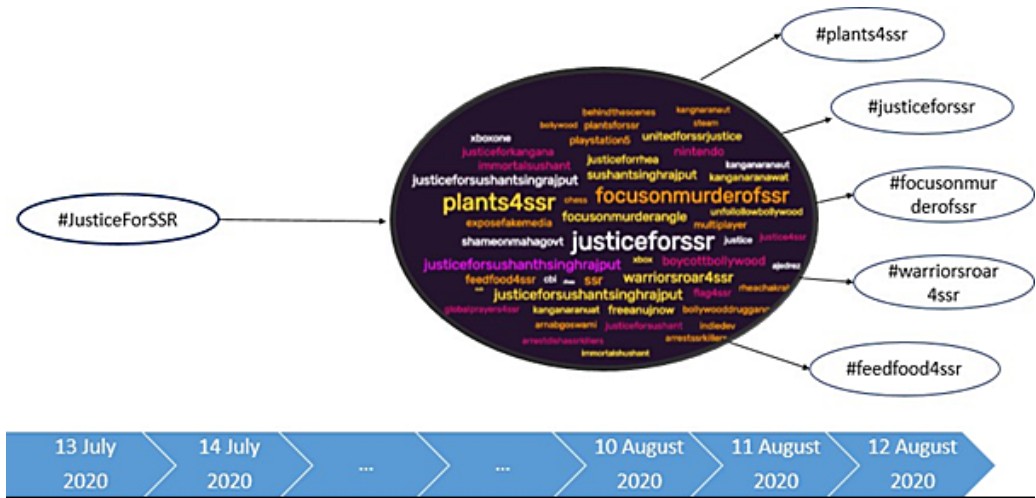

**Figure 11.** Viral Hashtags detection using initial hashtag #JusticeForSSR.

## 7. Conclusions

This paper presents a novel approach to detecting and understanding sentiment trends and emotional patterns of social media users in a celebrity suicide case. The process of detecting the sentiments and emotions is illustrated and results for sentiments (positive, negative and neutral) and emotions (positive, negative, anxiety, anger, sad) are shown. The highest positive sentiment (51.2%) and positive emotion (63.1%) is reported for the #Plant4SSR hashtag, the highest negative sentiment (38.9%) reported is for #SSR and the highest negative emotion (70.2%) reported is for #JusticeForSSR. On the other hand, more sad posts (31.7%) were reported with the #JusticeForSSR. Moreover, viral hashtags are detected from the posts from the initial occurrence of hashtags on a timeline. For further study, the detected emotion patterns and the sentiment patterns might be used to analyze the machine learning models applied. The models can predict future viral hashtags as well as the emotional sentiment analysis from social media in a similar context. Unsupervised learning methods could be applied for sentiment and emotion analysis.

**Author Contributions:** Conceptualization, A.A.M. and J.G.R.; methodology, A.A.M. and J.G.R.; software, A.A.M. and J.G.R.; validation, A.A.M., J.G.R. and R.A.; formal analysis, A.A.M.; investigation, A.A.M.; resources, A.A.M.; data curation, A.A.M. and J.G.R.; writing—original draft preparation, A.A.M. and J.G.R.; writing—review and editing, A.A.M. and J.G.R.; visualization, A.A.M. and J.G.R.; supervision, R.A.; project administration, A.A.M. and R.A. All authors have read and agreed to the published version of the manuscript.

**Funding:** This research received no external funding.

**Institutional Review Board Statement:** Not applicable.

**Informed Consent Statement:** Not applicable.

**Data Availability Statement:** Data are available from the first author and can be shared with anyone upon reasonable request.

**Conflicts of Interest:** The authors declare no conflict of interest.

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
