# Peer review of "Detecting and Understanding Sentiment Trends and Emotion Patterns of Twitter Users—A Study on the Demise of a Bollywood Celebrity"

_2504-2289, doi:10.3390/bdcc6040129_

Round 1

Reviewer 1 Report (Previous Reviewer 1)

The authors have addressed all the comments. So, I, therefore, recommend this paper for acceptance. 

Author Response

Dear Reviewer, 

Thank you so much for your effort and time to go through the paper again. Thanks for all your comments that enhanced the quality of our paper for the journal. As you mentioned minor spell check is required, we have covered the full paper once again and updated if any changes were required. 

Thank you.

Reviewer 2 Report (Previous Reviewer 2)

This version of your paper is much improved. The purpose and contribution of your research is now easier to follow. There is now good alignment among the title, abstract and article.

You have addressed the issues raised in the previous review, and framed the research more appropriately.

I have some minor suggestions for you to consider implementing if/when you submit a camera-ready version.

1. Line 243  Use a different font for the function.

2. Figure 1    Consider making the graphic wider which will enable you to avoid having to define abbreviated terms. However, if you keep the current graphic, you need to also add the full form for FIR

3. Line 247   is been illustrated - - > is illustrated

Author Response

Dear Reviewer, 

Thank you so much for your effort and time behind this paper. We appreciate all your valuable comments which enhanced the quality of this paper for the BDCC journal. 

All the minor comments are addressed, as below. 

  1. Line 243: Use the italic font for the function name.
  2. Figure 1: Add the full form of FIR in the figure name. Now it would be clear to understand the figure. 
  3. Line 247: Updated the sentence accordingly. 

We appreciate your overall comments on the paper. The paper is checked again, and if any style or spelling error is found, all were corrected accordingly. the cohesion of the "Introduction" section was checked to provide a background and necessary references are added accordingly. Secondly, to update the research design figures 2, 3, 4, 5, and 10 were described in detail.  Finally, the results of the work, with useful information including percentages of sentiment and emotion levels are added in the conclusion section. 

We have addressed all the minor and other comments as you mentioned.

Thank you.  

This manuscript is a resubmission of an earlier submission. The following is a list of the peer review reports and author responses from that submission.

Round 1

Reviewer 1 Report

This paper aims to analyze sentiment trends and emotion 53 patterns towards films and actors expressed by fan bases. Overall, the paper is easy to follow. However, while considering the paper as a whole, I found some major concerns about it which need to be addressed before it can be published.

1.     The abstract is short, descriptive, and lacks the motivation for the content of the manuscript to the most extent.

2.     The introduction can be reorganized and connectivity between its paragraphs—for instance, the transition from describing social networks such as Twitter to sentiment trends and emotional patterns.

3.     A common terminology should be established for the rest of the paper.

4.     The authors need to describe what is the key novelty about their paper and what is different in their work compared to other relevant papers.

5.       Figure 2 provide an overview of the proposed framework to detect sentiment trend and emotion pattern. However, it is unclear how those data were collected and why.

Reviewer 2 Report

Overall

Based on the title the purported aim is to detect and understand trends and emotion patterns of Twitter users based on a case study of the death of a Bollywood star.  The abstract explains that different sentiment analysis algorithms and LIWC were used. Hashtags were also investigated. However, I am at a loss to understand what this one case adds to the extant literature after reading the abstract. On page 4 in lines 139-145 contributions are listed. The novelty of the approach, however, is yet to be stated. I will read on as I am a reviewer; but as a reader, I would probably give up reading about now.

It does not appear that there is any breakthrough in performance of the algorithms. In lines 47-8 the authors state that the main motivation is to try to understand the psychology behind the posts and hashtags, but there is no discussion of psychological concepts. Apart from generalizations that negative news raises negative emotions and positive news raises positive emotions, I find it hard to understand how sentiment analysis will help us understand the underlying psychology.

The controversy caused by disinformation -- reporting a suicide as a murder -- is interesting and could help relate this one incident to other disinformation events. Framing the research as an investigation of the effect of disinformation on sentiment might be a way of finding a yet-to-be-filled research niche. The authors write “The general aim of this paper is therefore to analyze sentiment trends and emotion pattern towards films and actors expressed by fan-bases”. However, only one incident with one actor is investigated. Admitted SSR is a huge star, but that does not make the results generalizable.

The method is described adequately and the pseudocde makes the steps easy to examine. The methods seem standard and straightforward. The results are reported visually and in words although there is a lack of discussion.

Having read the whole article, I still do not know what was novel about the approach adopted (stated contribution 1) and do not understand the importance of the findings. The results of the experiment were reported with little to no discussion. Given the authors' claim to “understand the psychology”, there was little attempt to do so.

Major Issues

1. The novelty of the contribution to the literature needs to be explicitly stated not only so that reviewers can assess the claim, but to attract readers to read more than the abstract. The authors mention a “novel approach”, but do not let readers know what is novel.

2. The importance of the research needs to be established. What emotions people feel about a celebrity is not in itself of importance. Linking those emotions to events may be. However, the authors need to draw out this importance more explicitly.

3. In Lines 249-50, The regular feature extractors, such as, TF-IDF [26], n-gram models [27] etc. are not used. 

Please state the reason to help readers follow your decision process.

4. Discuss your results with reference to your research aim.

5. The conclusion needs to be more closely linked to the aims, results and discussion. The current conclusion could be used for any research investigating sentiment.

Minor issues

1. Line 16  Social media platforms (i.e., Facebook, Twitter, Instagram etc.)

i.e. is inappropriate. e.g. should be used as these are only examples of platforms and not the complete list of platforms

2. Lines 35-6 The emotions contained in the social media posts are expressed in various ways, one of which is through the contents of posted text messages

Listing the various ways would help readers less familiar with this topic.

3. Lines 44-46 Such user generated textual data in the posts is very much informal even so they contain deep insights into the personalities generating the posts

This reads as though it is surprising that informal language should give insights into personalities. My assumption would be that more formal texts would give fewer rather than more insights. This might be supported by the citation, but currently I suspect many may disagree. Consider rewording or hedging.

4. Line 65 Numbering of citations.

Normally citation numbers ascend rather than randomly jumping to [40] from [10]. If the PDF was created in LaTeX, I suspect that the authors used \bibitem{ } rather than a .bib file.

5. Line 112 misinformation

Misinformation is intentional while disinformation is unintentional. Reposting IMO may be both misinformation and disinformation.

6 Line 119-120 A recent paper surveys social media postings relating to covide-19 covers Twitter emotion research in detail

Grammar and typo need fixing.

7. Line 241 (shown in Algorithm 1)

There is no algorithm labelled 1. I assume that the Pseudocode is used in place of the algorithm.

8. authorit

Typo